# Second Opinion for Non-Surgical Root Canal Treatment Prognosis Using Machine Learning Models

**DOI:** 10.3390/diagnostics13172742

**Published:** 2023-08-23

**Authors:** Catalina Bennasar, Irene García, Yolanda Gonzalez-Cid, Francesc Pérez, Juan Jiménez

**Affiliations:** 1ADEMA, School of Dentistry, University of the Balearic Islands, 07122 Palma de Mallorca, Spain; c.bennasar@eua.edu.es; 2Department of Mathematical Sciences and Informatics, University of the Balearic Islands, 07120 Palma de Mallorca, Spain; irene.garcia@uib.es (I.G.); yolanda.gonzalez@uib.es (Y.G.-C.); 3Dental Public Health Service, IB-Salut, Balearic Islands, 07003 Palma de Mallorca, Spain; nidrell@gmail.com; 4TotIA Artificial Intelligence for Dentistry, 07006 Palma de Mallorca, Spain

**Keywords:** machine learning, outcome prediction, non-surgical root canal treatment, apical periodontitis

## Abstract

Although the association between risk factors and non-surgical root canal treatment (NSRCT) failure has been extensively studied, methods to predict the outcomes of NSRCT are in an early stage, and dentists currently make the treatment prognosis based mainly on their clinical experience. Since this involves different sources of error, we investigated the use of machine learning (ML) models as a second opinion to support the clinical decision on whether to perform NSRCT. We undertook a retrospective study of 119 confirmed and not previously treated Apical Periodontitis cases that received the same treatment by the same specialist. For each patient, we recorded the variables from a newly proposed data collection template and defined a binary outcome: Success if the lesion clears and failure otherwise. We conducted tests for detecting the association between the variables and the outcome and selected a set of variables as the initial inputs into four ML algorithms: Logistic Regression (LR), Random Forest (RF), Naive-Bayes (NB), and K Nearest Neighbors (KNN). According to our results, RF and KNN significantly improve (*p*-values < 0.05) the sensitivity and accuracy of the dentist’s treatment prognosis. Taking our results as a proof of concept, we conclude that future randomized clinical trials are worth designing to test the clinical utility of ML models as a second opinion for NSRCT prognosis.

## 1. Introduction

Apical Periodontitis (AP) is an inflammatory response caused by microorganisms in the root canal of an infected tooth, and its timely clinical treatment is the only viable alternative to undesired tooth loss. According to a recent systematic review and meta-analysis, half of the adult population worldwide has at least one tooth with AP, making it the most common endodontic disease among adults [1].

The conventional therapeutic option for the first-time treatment of AP is the NSRCT. Although NSRCT does not restore full functionality of the tooth (e.g., no innervation, no vascularization, no immune response), it maintains the tooth as well as the masticatory function and prevents infections. However, the treatment failure rate exceeds 15% after 5 years and 40% after 20 years [2].

Clinicians decide whether to perform NSRCT after estimating the prognosis of the treatment. This occurs after clinical and radiographic evaluation of the patient and is often based solely on the dentist’s own judgment, involving sources of error that can eventually lead to treatment failure. However, today dentists collect data about their patients, from demographics to clinical data, and the development of user-friendly software tools offers the possibility of extracting valuable information from these databases. Therefore, our research question can be summarized as follows: If a dentist frequently performs NSRCT and follows the same pre and intraoperative procedures, could this dentist improve the accuracy of the prognosis using his clinic’s database and machine learning algorithms as a second opinion?

Given the significant interest in the decision-making process of performing an NSRCT, the association between the outcome of treatment and different risk factors has been widely studied. These factors might be classified into two main groups: (1) patient-centered preoperative factors, such as pre-operative pain, systemic diseases (e.g., diabetes mellitus or cardiovascular diseases), tooth type (e.g., incisor, canine, premolar, or molar), and lesion size, and (2) the techniques and endodontic materials employed during the treatment and the type of reconstruction afterwards [2,3,4].

However, detecting the association between risk factors and outcomes is only the first step to advance in the development of accurate predictive methods: Association does not imply causation, and the predictive utility of the factors associated with the outcome must be tested. In general terms, this implies implementing a reliable predictive method that uses risk factors as input variables and generates an output that must be contrasted with the observed outcome. There are multiple options, but it can reasonably be argued that the more diverse the areas in which the forecasting method has shown efficacy, the greater the confidence in our conclusion about the predictive value of the factors. In this regard, ML algorithms are straightforward to implement and interpret and have been successfully applied to various classification and prediction problems, not only in dentistry [5,6,7,8,9,10], but also in many other areas [11,12,13,14,15].

Two recent studies have shown that some factors significantly associated (*p*-values < 0.05) with endodontic treatment outcomes have limited predictive power. In these studies, teeth-, patient-, and treatment-centered risk factors were used as covariates in ML algorithms to predict the outcome, and results showed poor performance on the testing set [16,17]. This example illustrates that association is a necessary but not a sufficient condition for causation. Variables can be diagnostically valuable and be associated with the outcome but still have low predictive power. In fact, the performance of a ML model in which all variables associated with the outcome are included might underperform compared to a judiciously chosen subset of variables. This highlights why variable selection procedures should be considered to optimize the performance of ML models to predicting the outcome of NSRCTs.

We build on a recently published article proposing a data collection template (DCT) for reporting endodontic outcome studies [18]. In this template, 38 preoperative variables were grouped into eight domains, including demographic data, patient medical history, clinical signs and symptoms, intraoral and extraoral examination, diagnostic data, radiographic techniques and findings, diagnosis, and the prognosis estimated by an expert dentist.

As proof of concept, we designed a retrospective study in which the DCT variables were recorded and those associated with the outcome were identified. Using four well-known ML models (LR, RF, NB, and KNN) we implemented a variable selection procedure to obtain a subset of associated variables with predictive value. The accuracy of these models in predicting the treatment outcome was evaluated and compared with the dentist’s prognosis. Results showed a significant improvement of the dentists’ prognosis when ML models were consulted as a second opinion.

## 2. Materials and Methods

### 2.1. Study Sample

A retrospective study was conducted, in which case histories of patients with AP who received NSRCT were randomly selected from the databases of a private clinic located in Mallorca, Spain, where patients were either scheduled for dental check-ups or were in an emergency. From this first group of cases, a further selection was made to include only those patients without any reported systemic disease, who received the treatment for the first time (not re-treatments), and whose records included:A general and dental clinical history with reports of general, facial, and oral inspection, as well as dental inspection, percussion and palpation;Results of a complementary thermal test with an ice pencil and periapical radiography; andAt least nine years of follow-up data for each patient, during which the dentist recorded the cases with a favorable (or unfavorable) recovery process towards recovery after performing the following procedure: a clinical examination measuring suppuration or functional incapacity and comparison of the diagnostic periapical radiography with a control one, to determine whether there had been a lessening in the lesion’s size.

The periapical X-rays were done with an X Mind Unity Acteon Satelec, with a focal point of 0.4 mm, at 70 Kv and 7 mA. Images were acquired with a Carestream 6100 digital X-ray sensor kit with an effective resolution of 15 LP/mm. All the X-rays were performed using the bisecting angle technique with a Rinn XCD (Dentsply) positioning system. Where a patient had suffered a fistula, fistulography with a gutta-percha point, Nº 25, was performed. According to their X-rays, we excluded patients from the study who were observed to have a vertical radicular fracture, or who did not have enough ferrule for the posterior reconstruction of the tooth.

Because of this filtering process, the number of patients finally included in the study dropped to 119. Patient consent was waived due to the lack of the possibility of identifying participating patients in the datasets. The Balearic Islands Research Ethics Committee (IB4015/19IP) granted approval for the study.

### 2.2. Intervention

The 119 patients with confirmed AP underwent identical endodontic treatment using the same materials administrated by the same endodontic specialist. Patients were put under a local anesthetic, and a rubber dam was placed over the treatment area, after which the pre-flaring of the coronal third, followed by the negotiation of the apical third, was undertaken. The working length was determined using a Morita Apex Locator and confirmed with a periapical X-ray. A K3 (Sybronendo) and a Protaper Gold (Dentsplay Maillefer) were utilized for canal negotiation and shaping, and manual metal instruments were used to give a conical form. Specifically, a 0.6 or 0.8 taper was used to prepare the canal, and “patency” was achieved by passively penetrating 0.5–1 mm beyond the apical terminus using a manual instrument No 8 or 10. In all cases, the working length employed was the radiographic apex. The coronal third was prepared mechanically, with EDTA (Ethylenediaminetetraacetic acid) as the irrigant, and sodium hypochlorite (5.25%) solution was used to prepare the middle and final thirds. The obturation of the canals was done using the warm vertical condensation method technique. The root canal sealer was AH Plus, made from epoxy-amine resin. The treated teeth did not show pores or over-obturation in the X-ray images after the procedure.

### 2.3. Variables and Outcome

For each patient, one author (CB, Endodontist with 30+ years of experience) evaluated the preoperative variables of 8 domains included in the DCT and estimated the prognosis before the analysis with ML models. The domains and variables are shown in Table 1. All measured variables contained categorical values, and the corresponding levels are also provided.

We defined the outcome both clinically and radiographically: Success occurs when both there are no symptoms or indications for further treatments, and the lesion disappears after NSRCT (Figure 1). Otherwise, failure occurs when either the clinical or radiographic outcome fails.

### 2.4. Statistical Analysis and ML Models

All statistical analyses and the deployment of the ML models were conducted using libraries from the open-source software, R. For the LR, RF, and NB models, the parameters and settings we used the default values provided by the R libraries for each method. For the KNN model, the number of neighbors used was k = 1. We first assessed the association between the variables and the outcome using the Pearson chi-square test or the Fisher exact test. After identifying the variables associated with the outcome, their predictive strength was tested using the four ML algorithms. For this purpose, we iteratively ran each ML model on the data using Leave-One-Out Cross Validation (LOOCV) and Backward Stepwise Selection (BSS) for variable selection [19].

LOOCV is a useful technique to evaluate the performance of any ML model, providing an unbiased estimate of the same. In short, the idea is to systematically omit a data point from the data set and use it as a validation set. The ML model is fitted on the remaining data set, and a predicted value is generated for the excluded observation. The process is repeated for each data point in the data set, and the predicted values of all data points are compared to the observed values to assess model performance. On the other hand, BSS is a variable selection technique used to identify the most relevant features for a predictive model. Starting with the full model that uses all the variables associated with the outcome, it considers the impact that the removal of each variable would have on the model’s performance and progressively removes, one at a time, variables that do not positively contribute to the model’s predictions/performance.

This LOOCV + BSS tuning procedure leads to the ML models that we ultimately run on the data, each of which contains a specific subset of the variables associated with the outcome. As for the metrics used to evaluate and compare the performance of the models, we computed the Sensitivity (Specificity) as the proportion of observed failures (successes) correctly predicted. We also computed the Positive Predictive Value (Negative Predictive Value) as the proportion of predicted failures (successes) that matched the observed failures (successes) and the Accuracy as the proportion of total true predictions.

## 3. Results

### 3.1. Association Analysis

Since all the DCT variables and outcome are categorical, the chi-square test or Fisher’s exact test was used to look for associations between them. As a result of these tests, we detected an association with the outcome in nine of the 38 DCT variables. In Table 2 we show their levels, *p*-values, and the effect size resulting from the association test. Once the variables associated with the outcome were identified, we conducted a correlation analysis (Spearman) to avoid potential collinearities, and our results indicate a low correlation between the different pairs of variables.

The first three listed variables were “Age”, “Highest Level of Education”, and “Arch”, which are demographic variables. In a second group of variables we find “Smoking” and “Patient Cooperation”, both from the domain Preoperative Data Related to the Patient (Clinical History). From the domain Preoperative Clinical Signs and Symptoms, we identified “Pain Relieved by” and “Time-lasting of the Pain”, while from the domain Preoperative Diagnosis we identified the variable “Periapical”. Finally, we detect an association between the outcome and the “Estimated Prognosis” from the dentist.

To quantify the association’s strength, we also computed the effect size for each variable. In general, these effects are moderate, and the variables with larger effect sizes were “Age” and “Highest level of education”.

### 3.2. Outcome Prediction

Once we identify the variables that show an association with the outcome, we use them as input variables in the LOOCV + BSS procedure described above. This is applied iteratively using each of the ML models separately, leaving us with a subset of variables for each. Specifically, the model requiring the lower number of covariates was NB, which only used the following five variables: “Highest educational level”, “Smoking”, “Patient cooperation”, “Time-lasting of the pain”, and “Prognosis”. In addition to these five variables, we have had to include the “Periapical” variable in KNN and RF, while LR forces us to include the “Age” variable in addition to the six aforementioned variables.

Results are summarized in Table 3, where we have included the dentist’s performance (DP) when assigning the prognosis level “Excellent” as the prediction of a successful outcome. We calculated the confusion matrix for each ML model, as well as for DP, and the values are displayed in Table 3 through the corresponding values of the True Positives (TP, number of failures predicted correctly), False Negatives (FN, number of failures predicted as successes), False Positives (FP, number of successes predicted as failures), True Negatives (TN, number of successes predicted correctly). Additionally, the values of each metric and its 95% confidence interval are shown for each ML model and for DP.

Overall, RF outperforms all other ML algorithms. However, as seen from the 95% confidence intervals in Table 3, the differences are not statistically significant, and we could say that all four ML models have the same performance. Instead, we found some statistically significant differences when comparing Sensitivity, NPV and Accuracy from the ML models and those of the dentist. Effectively, based on the values of TP and FN in Table 3, we compared the Sensitivity resulting from using the ML models with that of DP, and the differences were statistically significant for RF (*p*-value = 0.0076) and KNN (*p*-value = 0.025) while there were no significant differences with LR or NB (*p* values = 0.065). Using the values of FP and TN from Table 3, we did not find significant differences in the Specificity between ML models and DP, and we also did not find differences in the PPV when using TP and FP. We obtained significant differences in the NPV between DP and RF (*p*-value = 0.024), in the Accuracy between DP and RF (*p*-value = 0.005), and between DP and KNN (*p*-value = 0.027).

## 4. Discussion

The generation of knowledge and applications of Artificial Intelligence in various fields has experienced exponential growth in recent years. There are well-known reasons for this, such as the increase in accessible and high-quality data, the development of increasingly powerful hardware, and the creation of various user-friendly software capable of performing complex tasks.

According to the Stanford Institute Artificial Intelligence Index Report [20], worldwide private investment in AI in 2021 stood at approximately $93.5 billion, more than double the total private investment in 2020, while in 2020 there were four rounds of funding worth $500 million or more and in 2021 there were 15. Data management, processing, and cloud received the greatest amount of private AI investment in 2021—2.6 times the investment in 2020—followed by medical and healthcare. Regarding scientific publications, after growing only slightly from 2010 to 2015, the number of AI journal publications grew almost 2.5 times since 2015. As a percentage of all journal publications, AI journal publications in 2021 were about 2.5% of all publications, compared to 1.5% in 2010. Additionally, the number of patents filed in 2021 is more than 30 times higher than in 2015, showing a compound annual growth rate of 76.9%.

Of course, dentistry has not been unaffected by this intense process and AI has had a transformative impact on dental care delivery. It has improved diagnostic accuracy, enhanced treatment planning and working length determination, facilitated predictive analytics, detection and diagnosis of vertical root fractures, enhanced patient education, and expanded access to dental care through teledentistry [21,22,23,24,25,26,27,28,29,30]. As AI continues to advance, it holds the potential to further revolutionize dental care and improve patient outcomes.

One of the issues that seems of great practical interest to us is the potential use of the information collected from clinical cases to optimize decision-making processes. In this regard, we study the case of NSRCT and address two specific issues:Would a dentist who regularly performs NSRCT for AP cases, following the same protocol, using the same materials, and having a database where all the variables included in the DCT have been recorded, benefit from using ML algorithms as a second opinion on treatment prognosis?Would it be feasible to integrate this second opinion tool in a clinical setting?

To shed light on the answers to these questions, we implemented a proof-of-concept whereby a dentist, with more than 30 years of experience in performing NSRCT, evaluated the 38 variables of the DCT in a group of patients from her clinic. This patient sample deliberately did not include people with systemic diseases, which is a condition that could affect treatment outcome. Clearly, this exclusion criterion limits the generalizability of our results. However, including cases of patients with diabetes or cardiovascular diseases introduces a major complication in the study design. This proof-of-concept is a precursor to more intricate research designs we aim to embark upon, driven by our current study’s findings.

According to our results, a relatively small number of variables (9) of the DCT can be associated with the outcome, and the effect size was greater for variables Age and Highest level of education, both in the demographic domain.

In addition to detecting association, we investigated which of these variables have predictive power when used in a ML model. This is a necessary task that must be done to improve the performance of forecasting models, and it can be done in several ways. Specifically, we combine the LOOCV and BSS procedures with ML algorithms for variable selection. This resulted in the identification of a reduced set of variables, clearly illustrating that association does not imply prognostic value. It is worth mentioning that although the subset of predictors depends on the model, there is a kernel of five covariates that must be used in all of them: “Highest educational level”, “Smoking”, “Patient cooperation”, “Time lasting of the pain”, and “Prognosis”. There are also two non-predictor covariates for all the ML models: “Arch” and “Pain relieved by”, despite the fact that the effect size of one of them (“Pain relieved by”) is equal to that of a predictor variable (“Prognosis”), as can be seen in Table 2. Finally, the “Age” and “Periapical” variables only have prognostic utility in some of the ML models. For further analysis of the variables, we order them according to their importance in the model with the best performance (RF) and obtain that the most important is “Prognosis” (importance = 10.9), followed by “Highest educational level” (importance = 10.2), “Periapical” (importance = 8.6), “Smoking” (importance = 7.5), “Pain duration” (importance = 7), and “Patient cooperation” (importance = 3.6).

Aside from the dentist’s prognosis, over 66% (4 out of 6) of the variables with predictive value are either from the demographic domain or the patient’s medical history. The remaining predictors are from the Preoperative clinical signs and symptoms domain (16.7%) or from Preoperative diagnosis (16.7%). We did not find predictive value in any variable from the Preoperative clinical findings domain, Preoperative diagnostic data domain or Preoperative radiographic techniques and findings domain. This indicates that most of the variables with diagnostic value do not have prognostic value, while some variables without diagnostic utility are important in the prognosis.

Regarding the metrics used to evaluate the performance of the models, we used standard measures such as Sensitivity, Specificity, Positive Predictive Value, Negative Predictive Value, and Accuracy. After evaluating these metrics with the different models and comparing the corresponding confidence intervals, we did not find significant differences in the performance of the models. Likewise, we did not obtain significant differences in Specificity or PPV when comparing the prognosis of the dentist with that of the ML algorithms. However, we obtained significantly higher Sensitivity and Accuracy with two of the ML models (RF and KNN), as well as a significantly greater NPV with RF. This is because the numbers of TPs (FNs) identified by the algorithms are higher (lower) than that indicated by the dentist, while the numbers of FPs (TNs) indicated by the algorithms are lower (higher) than that provided by the dentist.

Accordingly, the dentist’s prognosis regarding lesion disappearance after NSRCT would be enhanced with the use of ML models, and therefore, the answer to the first question is yes.

The results that we have shown correspond to the application of the specific technique used by the operator, which was described in Section 2, and changing the technique used can reasonably be expected to lead to different results. While this is a point we will address in future work, we want to emphasize that the methodology to be followed will be like the one we have applied: (1) The dentist must register the DCT template for each patient treated with the new technique; (2) Conduct association tests; (3) Running the LOOCV + BSS procedure for variable selection; (4) Evaluate the performance of the ML models and use the best one as a second opinion for the prognosis of the treatment outcome.

Unlike AI algorithms based on the use of convolutional neural networks (i.e., Deep Learning), the ML algorithms we have applied can be trained on a standard laptop in a few minutes. The only requirement is that the dentist has their own database in a suitable format, such as an Excel sheet. In our opinion, this is a realistic option to implement at the clinical practice level which affirmatively answers the second question.

Our study has limitations and areas for improvement to increase generalizability. These include a larger sample size and the analysis of outcome prediction from multiple experts. Additionally, other ML models like Support Vector Machines or Neural Networks and a long-term follow-up on patients might also provide valuable insights. In our opinion, our results clearly indicate that all of these ideas are worth investigating, since progress in this direction could lead to the development of useful tools in clinical settings to optimize patient well-being.

## Figures and Tables

**Figure 1 diagnostics-13-02742-f001:**
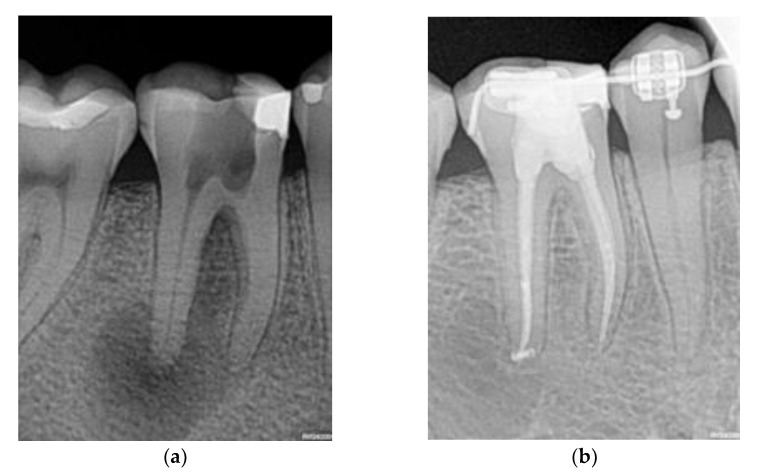
Example where the initial lesion (**a**) vanished after the NSRCT (**b**).

**Table 1 diagnostics-13-02742-t001:** Variables recorded preoperatively.

Domain	Variables
Demographic data	Gender (Male, Female, Other), Age (≤15, 15–24, 25–34, 35–44, 45–54, 55–64, ≥65), Highest level of education (Primary, Secondary, Post-secondary), Treated tooth number (1–32), Tooth type (Incisor, Canine, Premolar, Molar), Arch (Mandible, Maxilla)
Preoperative patient-related data (medical history)	ASA category, Allergies (No, Yes-Latex-Penicillin-Other), Premedication for endodontic treatment (Analgesic, Antibiotic, Other), Smoking (No, Everyday, Someday, Former), Recreational drugs/products (No, Everyday, Someday, Former), Patient co-operation (No, Yes), Anxiety (No, Yes), Sedation required (None, GA, IV, N_2_O:O_2_, Oral)
Preoperative clinical signs and symptoms	Spontaneous pain (No, Yes), Chronic pain in the orofacial region (No, Yes), Chronic pain outside the orofacial region (No, Yes), Pain triggered by (None, Sweet, Cold, Heat, Bite, Touch), Pain relieved by (None, Cold, Heat, Medication), Intensity of pain (Mild, Moderate, Severe), Time-lasting of the pain (Sec, Min, Continuous), Nature of pain (Sharp, Dull, Burning), Swelling (Absent, Present), Sinus tract (Absent, Present)
Preoperative clinical findings (intraoral and extraoral examination)	Soft tissue appearance (Normal, Abnormal), Lymphadenopathy (Absent, Present), Discoloration (No, Yes)
Preoperative diagnostic data (clinical)	Cold test (Negative, Positive), Percussion (Not sensitive, Sensitive), Palpation (Not tender, Tender)
Preoperative radiographic techniques and findings	Periapical index (No, PAI 1–2, PAI 3–5), Periapical rarefying osteitis (2–4 mm, 5–7 mm, ≥8 mm), Location of radiolucency (Apical, Furcal, Lateral), Canal curvature (<10°, 10°–30°, >30°)
Preoperative diagnosis	Pulp (Normal, Reversible pulpitis, Asymptomatic irreversible pulpitis, Symptomatic irreversible, Necrosis), Periapical (Normal, Asymptomatic AP, Symptomatic AP, Chronic Apical Abscess, Acute Apical Abscess), Number of roots
Estimated prognosis	Prognosis (Hopeless, Questionable, Fair, Good, Excellent)

**Table 2 diagnostics-13-02742-t002:** Variables associated with the outcome.

Variable	Levels	*p*-Value	Effect Size
Age	15–24; 25–34; 35–44; 45–54; 55–64; ≥65	0.0056	0.372
Highest level of education	Primary; Secondary; Post secondary	0.0016	0.33
Arch	Mandible; Maxilla	0.02	0.21
Smoking	No; Everyday; Someday; Former	0.046	0.26
Patient co-operation	No; Yes	0.028	0.21
Pain relieved by	None; Cold; Medication	0.003	0.31
Time-lasting of the pain	Sec; Min; Continuous	0.027	0.245
Periapical	Asymptomatic AP; Symptomatic AP; Chronic Apical Abscess; Acute Apical Abscess	0.01	0.31
Estimated Prognosis by clinician	Hopeless; Questionable; Fair; Good; Excellent	0.034	0.29

**Table 3 diagnostics-13-02742-t003:** Performance of the ML algorithms and the Dentist Prognosis.

Metric	DP	LR	RF	NB	KNN
TP	42	53	57	53	55
FN	27	16	12	16	14
FP	21	17	15	20	17
TN	29	33	35	30	33
Sensitivity	0.61 [0.48, 0.72]	0.77 [0.65, 0.86]	0.83 [0.72, 0.91]	0.77 [0.65, 0.86]	0.8 [0.68, 0.88]
Specificity	0.58 [0.43, 0.72]	0.66 [0.51, 0.79]	0.7 [0.55, 0.82]	0.6 [0.45, 0.74]	0.66 [0.51, 0.79]
PPV	0.67 [0.54, 0.78]	0.77 [0.65, 0.86]	0.79 [0.68, 0.88]	0.73 [0.61, 0.82]	0.76 [0.65, 0.86]
NPV	0.52 [0.38, 0.65]	0.67 [0.52, 0.8]	0.74 [0.6, 0.86]	0.65 [0.5, 0.79]	0.7 [0.55, 0.83]
Accuracy	0.6 [0.5, 0.69]	0.72 [0.63, 0.8]	0.77 [0.69, 0.84]	0.7 [0.61, 0.78]	0.74 [0.65, 0.82]

## Data Availability

Full datasets and R scripts are available upon reasonable request to the corresponding author.

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
