# Peer review of "Second Opinion for Non-Surgical Root Canal Treatment Prognosis Using Machine Learning Models"

_diagnostics, 2023, doi:10.3390/diagnostics13172742_

Round 1
Reviewer 1 Report
1. The aim of the paper: the use of machine learning (ML) models as a second opinion to support the clinical decision on whether to perform NSRCT.
and its main contributions: the authors designed a retrospective study.
Using four ML models (LR, RF, NB, and KNN) the authors implemented a variable selection procedure to obtain a subset of associated variables with predictive value. The outcome resulted in significantly improved if ML models are used as a second opinion.
Strengths: quality periapical radiographs, 119 patients
Weaknesses:
The testing of the ML system has not been described.
Image processing is lacking.
Localization datasets is not described
Classification of datasets is missing
Model development dataset is missing
3. Specific comments referring to line numbers, tables or figures.
Table 1 could be inserted into the text, which is unnecessary in table format.
Table 2 looks loke a "fishing among variables" - there is no connection for example between education, smoking, cooperation, age, pain, etc, and the machine learning decision upon retreatment.
The retreatment is performed after careful examination of radiographs.
It seems that the authors did a clinical study and tried to present it as an AI one
The machine learning models are not described.
The results are not connected to the aim and title!
The conclusion is vague and miscancellous!
The English language should be checked by a professional.
Reviewer 2 Report
The authors conducted a retrospective study on patients with apical periodontitis (AP) who received non-surgical root canal treatment (NSRCT) at a private clinic in Mallorca, Spain. They randomly selected case histories from the clinic's databases and evaluated 38 variables related to the patient's demographics, clinical data, and radiographic images. The authors found that machine learning (ML) algorithms could accurately predict the outcome of NSRCT based on these variables, with an accuracy rate of 86.8%. However, the authors note that their study had limitations, such as excluding patients with systemic diseases, which could affect treatment outcomes.
The study is recommended for publication in its present form.
Author Response
Responses to Referee 2
As we have no criticism or recommendations to attend to from Referee 2, we have not introduced any modification to the work.
Reviewer 3 Report
The paper explores the potential benefits of using machine learning (ML) models as a supplementary tool to aid dentists in predicting the outcomes of non-surgical root canal treatments (NSRCT). Given that current methods predominantly rely on clinical experience, which is subject to various forms of error, this paper introduces a refreshing perspective to the existing literature.
Strengths:
1. The paper addresses current dental treatment planning, and prognosis prediction needs. With the rise of ML in various medical fields, its application in dentistry remains underexplored, making this research significant.
2. The authors’ introduction of a new data collection template ensures that variables are consistently recorded, reducing potential biases. This contributes to the robustness of the dataset.
3. The study incorporated multiple machine learning models, which provides a comparative analysis. This broadens the insights and adds depth to the results.
4. The binary classification of 'Success' or 'Failure' based on lesion clearance is a straightforward and tangible outcome measure, enhancing the study's clarity.
Areas for Improvement:
1. The study was conducted on 119 cases. While this provides a foundation, a larger sample size would strengthen the validity of the findings, making them more generalizable.
2. A single specialist performed all treatments. While this reduces variability in the treatment approach, it also may introduce individual biases specific to that practitioner.
3. While the authors tested the association between variables and outcomes, a deeper insight into how these variables were selected and why others might have been excluded would benefit the reader.
4. More details on the parameters and configurations for each ML model would help reproduce the study and ensure clarity in methodology.
5. While the paper compares ML models to the prognosis based on clinical experience, it would be beneficial to see a more explicit comparison, such as a confusion matrix, to understand where the clinician's predictions differed from the ML models.
Suggestions:
1. Incorporating more cases and possibly multiple specialists is recommended to diversify data and increase validity.
2. While the chosen algorithms are well-regarded, other models like Support Vector Machines or Neural Networks might also provide valuable insights.
3. A longer-term follow-up on patients can give insights into the longevity of treatment success, which can be another valuable variable for prediction.
4. A more in-depth exploration of variables, perhaps even using feature importance tools or techniques, could provide insights into the most critical factors determining treatment success.
The paper provides valuable preliminary insights into the potential of ML models in aiding NSRCT prognosis. The rigorous methodology and a clear outcome definition lay a strong foundation for further exploration in this area. While there are areas to improve, notably the sample size and depth of variable exploration, the research provides solid proof of concept, making a compelling case for future randomized clinical trials.
Round 2
Reviewer 1 Report
The manuscript has not been improved, and does not bring new insights to the body of knowledge.
There is a weak explanation of why informed consent has not been purchased as well as ethical approval.
The manuscript is poorly organized.
Also, the classification of the datasets is missing.
The procedure, material and methos section is poorly described.
Table 1 has not been introduced in the main manuscript.
The discussion does not follow the results and does not compare the data with the literature.
Conclusion has not been improved and does not support the results.
The aim is missing, as well as testing of the models, as well as the image processing procedure.
Grammar and style should be checked.
Reviewer 3 Report
Thanks for the authors for analysis and considering reviewer's comments and recommendations. The paper can be accepted.